



# **Dust pollution in China affected by different spatial and**
**temporal types of El Niño**
Yang Yang[1*#], Liangying Zeng[1#], Hailong Wang[2], Pinya Wang[1], Hong Liao[1]
[1]Jiangsu Key Laboratory of Atmospheric Environment Monitoring and Pollution
Control, Jiangsu Collaborative Innovation Center of Atmospheric Environment and
Equipment Technology, School of Environmental Science and Engineering, Nanjing
University of Information Science and Technology, Nanjing, Jiangsu, China
[2]Atmospheric Sciences and Global Change Division, Pacific Northwest National
Laboratory, Richland, Washington, USA
*Correspondence to yang.yang@nuist.edu.cn
[#] These authors contributed equally to this work





**Abstract**
Dust is an important aerosol affecting air quality in China in winter and spring that
is potentially influenced by the interannual climate variability associated with El Niño.
Here, the impacts of El Niño with different temporal and spatial types on dust pollution
in winter and spring in China and the potential mechanisms are investigated using an
state-of-the-art earth system model (E3SMv1). We find that the Eastern Pacific (EP)
and Central Pacific (CP) El Niño both increase wintertime dust concentrations by 5–50
µg m$^{-3}$ over central-eastern China. Due to a stronger wind and lower relative humidity,
which favor dust emissions near sources, and a strengthened northwesterly and reduced
precipitation, which are conducive to dust transport, dust concentrations during the CP
El Niño are 5–20 µg m$^{-3}$ higher in northern China than during the EP El Niño. El Niño
with a short duration (SD) increases winter dust concentrations by 20–100 µg m$^{-3}$ over
northern China relative to the climatological mean, while there is a decrease of 5–50
µg m$^{-3}$ during the long duration (LD) El Niño, which are also related to the El Niño-
induced changes in atmospheric circulation, precipitation, and relative humidity. In the
following spring season, all types of El Niño events enhance dust over the northern
China, but only the increase during the LD El Niño is statistically significant,
suggesting that the weaker intensity but longer duration of the LD El Niño events can
significantly affect spring dust in China. Our results contribute the current knowledge
of the influence of El Niño on dust pollution, which have profound implications for air
pollution control and dust storm prediction.



## 1. Introduction


Dust, one of the most important types of natural aerosols, has significant impacts
on Earth's radiative balance (Seinfeld et al., 2004), regional and global climate (Kok et
al, 2018; Yang et al.,2017), the hydrological cycle (Huang et al., 2014), agricultural
production (Sivakumar, 2005), public health and transportation activities (Goudie,
2014). The Gobi Desert and the Taklamakan Desert in northwestern China are
important contributors to dust concentrations in East Asia and even globally, and about
30% of the dust from the sources in China can be transported to the downwind areas
over long distances (Chen et al., 2017). Despite China's vigorous efforts to combat
desertification since the beginning of $21^{st}$ century, strong and widespread dust storms
still occurred in China in recent years (Yin et al., 2021). Therefore, a deeper and more
scientific comprehension of the factors affecting dust aerosols in China is urgently
needed for the early warning and mitigation of dust pollution.
In recent years, the influence of meteorological conditions on dust pollution in
China has attracted considerable attention (Guo et al., 2019; Li et al., 2020; Lou et al.,
2016; Shi et al., 2021; Yin et al., 2021; Zhu et al., 2008). Under global warming in
recent decades, dust emissions and the frequency of dust storms in northern China
decreased (Shi et al., 2021), which was attributed to the reduced frequency and intensity
of Mongolian cyclones, related to the weakened westerly jet stream and atmospheric
pressure in northern China and Mongolia, in a warming climate (Zhu et al., 2008). Due
to a combination of changes in disruptive temperature anomalies in the Mongolian dust
source region, the occurrence of super Mongolian cyclone, and the anomalies of sea ice
in the Barents and Kara Sea and sea surface temperature (SST) in the east Pacific and
northwest Atlantic, China experienced the strongest dust pollution in spring 2021(Yin
et al., 2021). Lou et al. (2016) pointed out that springtime dust concentrations exhibited
a significant negative correlation with the East Asian Monsoon Index over most of
China with a correlation coefficient of –0.64 in their model simulations, and they found
that anomalous northwesterly winds in weak East Asian monsoon years led to a strong
dust transport from Mongolia to China. Mao et al. (2011) illustrated that the negative
(positive) phase of Arctic Oscillation (AO) can lead to an increase (decrease) in the
frequency of dust storms in northern China due to the increase (decrease) in the
frequency of cold air outbreak over Mongolia.
El Niño-Southern Oscillation (ENSO) is a well-known mode of climate variability



generated by coupled ocean-atmosphere interactions that can exert a far-reaching
impact on global climate despite its origin in the tropical Pacific Ocean (Trenberth,
1997; Yang et al., 2016a, 2016b; Zeng et al., 2021). Numerous studies have
demonstrated that El Niño can affect dust emission and transport by modulating large-
scale atmospheric circulation, precipitation and temperature (Lee et al., 2015; Li et al.,
2021). Using observational data over 1961–2002, Lee et al. (2015) found that the under
the negative AO phase, frequency of spring dust events in northern China during El
Niño was 30% higher than that during La Niña years. Li et al. (2021) used dust surface
concentration data (1982–2019) from MERRA-2 reanalysis to study the impacts of
ENSO events on global atmospheric dust loading and found that dust concentrations
were positively correlated with Southern Oscillation Index (SOI, a consistently
negative SOI is El Niño and the opposite is La Niña) over northwestern China, which
suggests that El Niño was associated with a decrease in dust concentrations. Modeling
studies driven by reanalysis data also revealed a relatively weak positive relationship
between SOI and dust emissions over Gobi Desert, although this correlation has a large
spatiotemporal variation (Gong et al.,2006; Hara et al.,2006). These numerical studies
used regional models driven by or nudged to reanalysis meteorological fields, which
could be influenced by factors other than El Niño. Recent studies have indicated that
the El Niño impact on air pollutants can be better represented by the superposed SST
perturbation method (Yu et al., 2019; Zhao et al., 2018; Zeng et al., 2021), considering
the influence of ENSO alone. To the best of our knowledge, no study has yet used this
approach to investigate the relationship between El Niño and dust pollution in China.

Additionally, previous studies mainly focused on the influences of general El Niño

on dust over China, while El Niño can be classified into different temporal types (e.g.,
short duration (SD) and long duration (LD) El Niño; Guo and Tan, 2018) and spatial
types (e.g., East Pacific (EP) and Central Pacific (CP) El Niño; Kao and Yu, 2009).
During different spatial and temporal types of El Niño, patterns of precipitation and
atmospheric circulation are also different in China (Yu et al., 2019; Zeng et al., 2021),
and they could have distinct effects on wintertime and springtime dust pollution in
China. Nevertheless, most of the existing studies have focused on the effects of various
spatial and temporal types of El Niño events on anthropogenic aerosols, while few
studies have examined their effects on natural aerosols, such as dust, and their
associated mechanisms, which are crucial for predicting and combating dust pollution
in the near future.





In this work, the effects of different spatial and temporal types of El Niño on
wintertime and springtime dust pollution in China and the mechanisms behind the
impacts are examined using the Energy Exascale Earth System Model version 1
(E3SMv1). The methods and model description are described in Section 2. The
quantitative impacts of various temporal and spatial types of El Niño events on
wintertime and springtime dust concentrations in China and the associated mechanisms
are elaborated in Section 3. Section 4 summarizes the key results and conclusions of
the study.

## 2. Data and Methods

### 2.1 Data

Global SST patterns and SST anomalies during El Niño events of different
temporal and spatial types are constructed using the merged Hadley-NOAA/OI dataset
which has a horizontal resolution of $1° \times 1°$ from 1870 to 2017 (Hurrell et al., 2008).
The monthly ERA5 reanalysis data (Hersbach et al., 2020) are applied to evaluate the
simulated meteorological parameters during El Niño events.
Hourly observations of $PM_{10}$ (particulate matter less than 10 μm in diameter)
concentrations in China from 2015 to 2021 derived from the China National
Environmental onitoring Centre (CNEMC) and the Deep Blue aerosol products
(Platnick et al., 2015) from Moderate Resolution Imaging Spectroradiometer (MODIS)
on Terra satellite, including monthly Aerosol Optical Depth (AOD) at 550 nm and the
Ångström exponent (α) from 2001–2020, are applied to evaluate the performance of
dust simulation in the model. The satellite dust optical depth (DOD) is calculated
following Yu et al. (2021).

### 2.2 El Niño events identified as different spatial and temporal types

We first clarify the definition of different temporal and spatial types of El Niño
events here. The notation of $year^0$ is used to denote the first year of El Niño
development, and $Jan^0$, $Feb^0$, ..., and $Dec^0$ indicate the individual months of that year,
while $year^{1,2,\cdots}$ and $Jan^{1,2,\cdots}$, $Feb^{1,2,\cdots}$, ...., and $Dec^{1,2,\cdots}$, respectively, denote the
following years and months therein. Niño 3.4 index is defined as the anomaly of
detrended SST in the Niño 3.4 region (170°W–120°W, 5°S–5°N). Niño 3/4 index
($I_{Niño3}/I_{Niño4}$) is same as Niño 3.4 index, but in the Niño 3/4 region (150°W–90°W, 5°S–
5°N; 160°E–150°W, 5°S–5°N).





For the classification of different temporal types, following Wu et al. (2019), El
Niño events are firstly selected if any of 3-month running averaged Niño 3.4 index
during Oct[0]–Feb[1] greater than 0.75°C . Then the LD El Niño event is identified once
any of Niño 3.4 index during Oct[1]–Feb[2] is higher than 0.5°C; otherwise, it is a SD El
Niño event.
Following Yu et al. (2019), the El Niño events, selected with 3-month running
averaged Niño 3.4 indices higher than 0.5°C for five consecutive months, are classified
into different spatial types based on the EP El Niño index ($I_{EP}$) and the CP El Niño index
($I_{CP}$). The definition of these indices is given below.
$$I_{EP} = I_{Niño3} - \alpha \times I_{Niño4} \tag{1}$$
$$I_{CP} = I_{Niño4} - \alpha \times I_{Niño3} \tag{2}$$
$$\alpha = \begin{cases} 0.4, & I_{Niño3} \times I_{Niño4} > 0 \\ 0 , & I_{Niño3} \times I_{Niño4} \leq 0 \end{cases} \tag{3}$$
If the mean $I_{EP}$ is greater than the $I_{CP}$ during Oct[0]–Feb[1] of an El Niño, then it is an
EP El Niño event; else, it is a CP El Niño event.
The time series of Niño 3.4 index derived from Hadley-NOAA/OI 1870–2017 data
is shown in Figure S1. Using the definitions described above, for El Niño with different
temporal types, 22 SD El Niño events and 8 LD ones are extracted during this time
period; for El Niño with different spatial types, 26 EP El Niño events and 8 CP ones are
extracted.
**2.3 Model description and experimental design**
To investigate the impacts of El Niño of different spatial and temporal types on
dust aerosol in China, this study utilizes the U.S. Department of Energy (DOE)
E3SMv1 (Golaz et al., 2019). As a model developed from the well-known CESM1
(Community Earth System Model version 1), E3SMv1 provides significant
improvements to the atmospheric component, including processes associated with
aerosol, cloud, turbulence, and chemistry (Rasch et al., 2019). We choose the horizontal
resolution of about 1° and 30 vertical layers. E3SMv1 predicts aerosols including
mineral dust, sea salt, sulfate, primary and secondary organic aerosols, and black carbon
in the four-mode Modal Aerosol Module (MAM4) (Wang et al., 2020). E3SMv1
represents dust-related processes in the atmosphere and land model components (Feng
et al., 2022). Dust emissions are calculated at each model time step according to the
wind erosion dust scheme proposed by Zender et al. (2003), which is related to 10-
meter wind speed, surface soil moisture content, soil erodibility, vegetation cover and





threshold friction velocity.

The following simulations are performed. A "CLIM" experiment applying the

prescribed climatological mean of monthly SST during 1870–2017 is integrated for 30
years. Four sets of sensitivity simulations, "SD", "LD", "EP" and "CP", are driven by
the monthly SST representing the composite of SD, LD, EP and CP El Niño events,
respectively, which is generated through adding the mean monthly SST anomalies from
Jul[0] to Jun[1] of the SD, LD, EP, and CP El Niño events (Fig. S1), respectively, to the
climatological SST between 60°S and 60°N. All the sensitivity experiments have 3
ensemble members with diverse initial conditions branched from different years of the
CLIM simulation. All sensitivity experiments are run for 13 years with the last 10 years
used for analysis. The differences of model fields between the sensitivity simulations
and CLIM represent the influences of El Niño events with different spatial and temporal
types on dust aerosols. All other external factors such as greenhouse gas concentrations,
insolation, anthropogenic aerosols and their precursor emissions are hold at present-day
conditions (year 2014). The SST anomalies relative to the 1870–2017 climatology
during SD, LD, EP and CP El Niño events are shown in Fig. 1.

**2.3 Model evaluation**

To evaluate the model performance in dust simulation, we compare the simulated

near-surface dust concentration and dust optical depth (DOD) over China with observed
$PM_{10}$ concentrations and satellite retrieved DOD, respectively. The model can
reproduce the spatial distribution of springtime dust in China, with high dust
concentrations in northwestern China and low in southern and northeastern China (Fig.
S2). The spatial correlation coefficient between the simulated dust concentrations in
E3SMv1 and observed near-surface $PM_{10}$ concentrations is +0.55. However, the model
strongly overestimates dust concentrations over the source regions, which were also
reported in many previous studies using the E3SMv1 and CESM (the predecessor of
E3SMv1) (Wang et al., 2020; Wu et al., 2019). The high model bias near the sources is
also confirmed by comparing DOD between model simulation and satellite retrieval. It
suggests that the dust emissions are overestimated in northwestern China in the model.
However, we also note that the E3SMv1 underestimates the transport of dust from
source regions (Wu et al., 2020; Feng et al, 2022), thus the dust over eastern China is
comparable to observations.



## 3. Results

### 3.1 Impacts of different El Niño types on winter dust pollution

The simulated effects of the four types of El Niño with different spatial positions (EP and CP) and durations (SD and LD) on the DJF ground-level dust concentrations are shown in Fig. 2. As for different spatial types of El Niño events, the effects on DJF dust concentrations in China are similar, with an increase in dust concentrations of 5–50 µg m$^{-3}$ over central-eastern China during EP and CP El Niño compared to the climatological means. The spatial pattern of dust changes is consistent with previous modeling studies (Lee et al., 2015; Li et al., 2021). Although the influences of EP and CP El Niño on the DJF dust concentrations resemble each other in the spatial patterns over China, the magnitudes of the influences are different. During CP El Niño relative to the climatological mean, dust concentrations increase more significantly over central-eastern China, with the increases of 20–50 µg m$^{-3}$, 5–20 µg m$^{-3}$ higher than that during EP El Niño. The large increase during CP El Niño is also more widespread than that during EP El Niño.

As for different temporal types of El Niño events, their effects on DJF dust concentrations over China are quite different. SD El Niño events cause an increase in DJF near-surface dust concentrations of 20–100 µg m$^{-3}$ in northern China and about 5–20 µg m$^{-3}$ in southern China. Whereas during LD El Niño events, winter dust concentrations have a decrease of about 5–50 µg m$^{-3}$ in northern and northeastern China relative to the climatology and no significant change is shown in southern China. In contrast to LD El Niño events, SD El Niño events have positive DJF dust concentration anomalies of 5–20 µg m$^{-3}$ in southern China and a maximum over 100 µg m$^{-3}$ in northern China and the Gobi Desert. Furthermore, DJF dust concentrations over the Taklamakan Desert, one of the largest dust sources in China, have an increase during LD El Niño events and an insignificant decrease during SD El Niño events.

Overall, these changes in dust concentrations indicate that CP El Niño events have stronger and more widespread impacts on DJF dust concentrations than EP El Niño, and the SD and LD El Niño events exert opposite impacts on DJF dust in China.

### 3.2 Mechanisms of the different El Niño impacts on winter dust

Meteorological factors such as 10-m wind speed, relative humidity and atmospheric circulation play a dominant role in altering dust concentrations by altering emissions, atmospheric transport, and wet scavenging of dust (Csavina et al., 2014).



Dust changes are also controlled by the El Niño-related changes in atmospheric
circulation and precipitation (Gong et al., 2006; Hara et al., 2006). The 10-m wind speed,
atmospheric circulation, relative humidity, precipitation anomalies, and related
processes during EP, CP, SD and LD El Niño are investigated here to reveal the
mechanisms of the influence of the four types of El Niño on dust over China.
During the CP, EP, and SD El Niño, DJF mean 10-m wind speed increases in the
Gobi Desert and northwestern China compared to the climatological mean (Fig. 3),
which favors the local dust emission over these regions. Whereas for the LD El Niño
event, the positive 10-m wind speed anomaly is greatly weakened, compared to the
other three types of El Niño events, and negative 10-m wind speed anomalies are
triggered in the Gobi Desert and northern China (Fig. 3e), which is not conducive to
dust emission during the LD El Niño events. The CP El Niño events trigger stronger
positive 10-m wind speed anomalies (0.1–0.3 m s$^{-1}$) than the EP El Niño events over
the Gobi Desert and northern China (Fig. 3c), which could lead to a greater local dust
emission. Compare to the LD El Niño, SD El Niño events produce significant positive
10-m wind speed anomalies of approximately 0.3 m s$^{-1}$ in the Gobi Desert and northern
China (Fig. 3f), which is consistent with the increase/decrease in local DJF dust
concentrations during the SD/LD El Niño (Fig. 2). This suggests the importance of 10-
m wind speed in the dust changes during the El Niño events in China.
Figure 4 shows the atmospheric circulation anomalies for the four El Niño events.
All types of El Niño have negative anomalies of sea level pressure (SLP) in central-
eastern China, except the LD El Niño that shows a negligible SLP change in winter.
Meanwhile, during the EP, CP, and SD El Niño events, anomalous Mongolian cyclone
can strengthen the local ascending flow to lift more dust particles into the free
atmosphere. The anomalous northwesterly during CP and SD El Niño (Figs. 3b and 3d)
can transport these dust aerosols to central-eastern China, leading to the strong increases
in dust concentrations there (Figs. 2b and 2d). While during the LD El Niño, the lower
atmosphere in the Gobi Desert and northern China is controlled by a weak anomalous
high pressure accompanied by anomalous southeasterly that weakens the prevailing
northwesterly in winter and hinders the vertical lifting and southward transport of dust.
Our previous work has confirmed the ability of E3SM in reproducing the
atmospheric circulation in El Niño with different durations (Zeng et al, 2021). Here we
further evaluate the circulations in E3SM simulations during EP and CP El Niño events
by using ERA5 reanalysis data. The anomalous DJF mean 10-m wind speed and 850


hPa wind fields in the typical EP El Niño (2006/07) and CP El Niño (2014/15) relative
to the climatology (1950–2017) from ERA5 are presented in Fig. 5. Although the
increase in 10-m wind speed over northwestern China in the EP El Niño simulated in
the model is inconsistent with the ERA5 results, E3SM does capture the large increase
in wind speed over the Gobi Desert during the CP El Niño relative to the climatological
mean and EP El Niño. Moreover, the anomalies in wind fields during EP and CP El
Niño (i.e., anomalous southerly during EP El Niño and anomalous northwesterly during
CP El Niño) are well reproduced by E3SM. It suggests that the atmospheric circulation
features during different types of El Niño are captured by the model. Also note that the
observational differences can be induced by other climate factors besides El Niño,
leading to a potential inconsistency in El Niño impact between model and observation.
The effect of relative humidity (RH) on dust concentration is also essential,
considering that a decrease in RH leads to a decrease in the threshold friction velocity
at high RHs (>40%), which further enhances dust emission flux and atmospheric
concentration (Csavina et al., 2014). Both EP and CP El Niño events have negative
anomalies in DJF RH in the Gobi Desert (Figs. 6a and 6b). The decrease in RH reduces
the dust threshold friction velocity and favors dust emission from the Gobi Desert. The
CP El Niño produces more pronounced and widespread negative RH anomalies in the
Gobi Desert and northwestern China than the EP El Niño. It gives approximately 3%
stronger negative RH anomalies (Fig. 6c), resulting in stronger and more widespread
increases in DJF dust concentrations during the CP El Niño event (Fig. 2c). As for El
Niño with different duration, the SD El Niño leads to significant decreases in DJF RH
of about 3% near the south part of the Gobi Desert, while increases in RH are located
over north part of the Gobi Desert during the LD El Niño (Figs. 6g and 6j), likely
resulting in the opposite changes in dust emissions. The ERA5 reanalysis data also
show the same RH variations during the different spatial and temporal types of El Niño
as the E3SM simulations described above (Fig. S3). Among all four types of El Niño
events, RH anomalies are consistent with the distribution of dust concentration
anomalies, which indicates that RH plays an important role in affecting variations in
dust emissions and concentrations in China during El Niño.
Fig. 7 shows the simulated changes in DJF dust emissions during different El Niño
events. During the EP and CP El Niño, DJF dust emissions are enhanced in the Gobi
Desert and northwestern China relative to the climatological average. The dust emission
increase is larger during the CP El Niño than the EP El Niño, which is consistent with



the higher positive DJF dust concentration anomalies during the CP El Niño.
Furthermore, the SD El Niño causes a significant increase in dust emissions of about
0.5 g m$^{-2}$ d$^{-1}$ in the Gobi Desert compared to CLIM, while the LD El Niño causes a
decrease in dust emissions. These suggest that different types of El Niño events alter
the DJF dust emissions in China by changing the 10-m wind speed and RH, which is
the important cause of the variation in DJF dust concentrations in China.
Furthermore, a reduced DJF precipitation during both EP and CP El Niño events
weakens the wet removal of dust from the atmosphere in northern China (Fig. S4),
further enhancing the increases in DJF dust concentrations.
**3.3 Spring dust pollution affected by El Niño events**
The changes in near-surface dust concentrations over China in the following spring
during the decaying phase for different spatial and temporal types of El Niño are also
examined (Fig. 8). During the following spring, all El Niño events trigger large positive
anomalies of March-April-May (MAM) dust concentrations in northern China.
However, the increases in dust concentrations during the EP, CP and SD El Niño relative
to the climatological average fail the 90% significance test, indicating that the effects
of these types of El Niño events on the dust pollution in northern China in the following
spring are uncertain, likely related to the large internal variability of the climate system.
In contrast to the strong reduction in dust concentrations over the Gobi Desert and
northern China during the LD El Niño in DJF, the effect in MAM reverses to a
significant increase in dust concentrations over these regions by 50–100 μg m$^{-3}$ (Fig.
8e). It suggests that the weaker intensity but longer duration of LD El Niño than the SD
El Niño can significantly affect spring dust aerosols in China.
During LD El Niño events, MAM 10-m wind speed significantly increases over
the Gobi Desert (Fig. S5), which facilitates the local dust emissions, although RH only
shows an insignificant decrease over the dust source region (Fig. S6). It can be
confirmed by the significant increases in MAM dust emissions by about 0.5 g m$^{-2}$ d$^{-1}$
over the Gobi Desert and northwestern China during LD El Niño events (Fig. 9). Then
the strengthened northwesterly brings more dust to northern China during LD El Niño
events (Fig. S7). Along the transport pathway, the weakened precipitation reduces the
dust wet removal (Fig. S8), leading to the strong increase in MAM dust concentration
over northern China during the LD El Niño.



## 4. Conclusion and discussions

Dust, as an important air pollutant affecting air quality in China in winter and spring, can be modulated by the interannual variations in El Niño-induced atmospheric circulation and precipitation anomalies. In this study, the state-of-the-art E3SM model is used to simulate the effects of different temporal types of El Niño events with short (SD) and long duration (LD) and different spatial locations of El Niño events with sea surface temperature anomalies located in Central Pacific (CP) and Eastern Pacific (EP) on dust concentrations in China.

Both CP and EP El Niño events cause 5–50 μg m$^{-3}$ positive anomalies in winter (DJF months) surface dust concentrations in central-eastern China. Compared to the EP El Nino, the CP El Nino triggers a stronger wind and negative RH anomalies that lead to greater local dust emissions. Then the anomalous northwesterly transports the dust aerosols to central-eastern China during the CP El Nino, accompanied by a reduced precipitation and wet removal of dust from the atmosphere, resulting in 5–20 μg m$^{-3}$ higher and more widespread DJF dust concentration increases in northern China. For the different temporal types of El Niño events, wind speed significantly increases over the Gobi Desert and northern China during the SD El Niño, favoring dust emissions. Meanwhile, the anomalous northwesterly can increase the transport of dust aerosols to central-eastern China, leading to an increase in DJF near-surface dust concentrations of 20–100 μg m$^{-3}$ in northern China and 5–20 μg m$^{-3}$ in southern China relative to the climatological mean. On the contrary, the LD El Niño reduces wind speed over the Gobi Desert and northern China, which weakens dust emissions, accompanied with the atmospheric circulation anomalies unfavorable for dust transport, leading to the DJF dust concentration decrease by 5–50 μg m$^{-3}$ in northern and northeastern China relative to the climatological mean.

In the following spring season, the four types of El Niño events with different durations and spatial positions all cause positive dust concentration anomalies in northern China. However, only the changes during the LD El Niño are statistically significant. This is mainly due to an increase in 10-m wind speed over the Gobi Desert during the LD El Niño, which enhances the local dust emissions, and then the strengthened northwesterly brings more dust to the northern China. At the same time, the weakened precipitation reduces the dust wet removal along the transport pathway. It suggests that the weaker intensity but longer duration of LD El Niño events than SD



El Niño can significantly affect dust aerosols in China in spring.

Our results contribute to the current knowledge of the vital influence of different

types of El Niño on dust pollution in winter and spring over China, which have
profound implications for air pollution control and dust storm prediction in China.
Notwithstanding, we also note that the E3SMv1 overestimates dust emissions from the
source regions and underestimates the long-range transport of dust (Wu et al., 2020;
Feng et al, 2022), which may lead to biases in the estimate of El Nino impact on dust
concentrations in China. In future studies, the influences of different types of La Niña,
the cooling phase of ENSO, on dust pollution in China, warrants further investigation.
Besides, other natural aerosols, such as sea salt, are also influenced by El Niño events,
which is not taken into account in this study. In addition to natural sources, dust in
China can also be from anthropogenic emissions (Chen et al., 2019; Xia et al., 2022),
and their relations with El Niño require further study.



*Code and data availability*
The E3SMv1 model is available at https://github.com/E3SM-Project/E3SM (last access:
25 Mar 2022) (http://doi.org/10.11578/E3SM/dc.20180418.36, E3SM project, 2018).
Our results can be made available upon request.

*Author contributions*
YY designed the research and analyzed the data. LZ performed the model simulations.
All the authors including HW, PW, and HL discussed the results and wrote the paper.

*Competing interests*
The authors declare that they have no conflict of interest.

*Acknowledgments*
HW acknowledges the support by the U.S. Department of Energy (DOE), Office of
Science, Office of Biological and Environmental Research (BER), as part of the Earth
and Environmental System Modeling program. The Pacific Northwest National
Laboratory (PNNL) is operated for DOE by the Battelle Memorial Institute under
contract DE-AC05-76RLO1830.

*Financial support*
This study was supported by the National Natural Science Foundation of China (grant
41975159), the National Key Research and Development Program of China (grant
2020YFA0607803 and 2019YFA0606800) and Jiangsu Science Fund for Distinguished
Young Scholars (grant BK20211541).



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

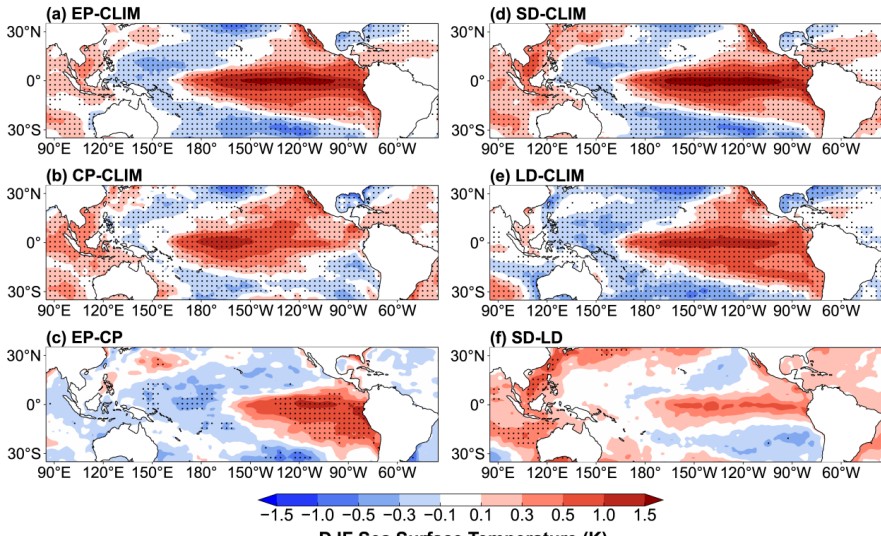


**Figure 1.** Composite differences in DJF mean SST (°C) between (a) EP, (b) CP, (d) SD, (e) LD El
Niño events and climatological mean over 1870–2017, and (c) between EP and CP, and (f) between
SD and LD El Niño events. Statistically significant differences at 98% from a two-tailed T-test are
stippled.

594





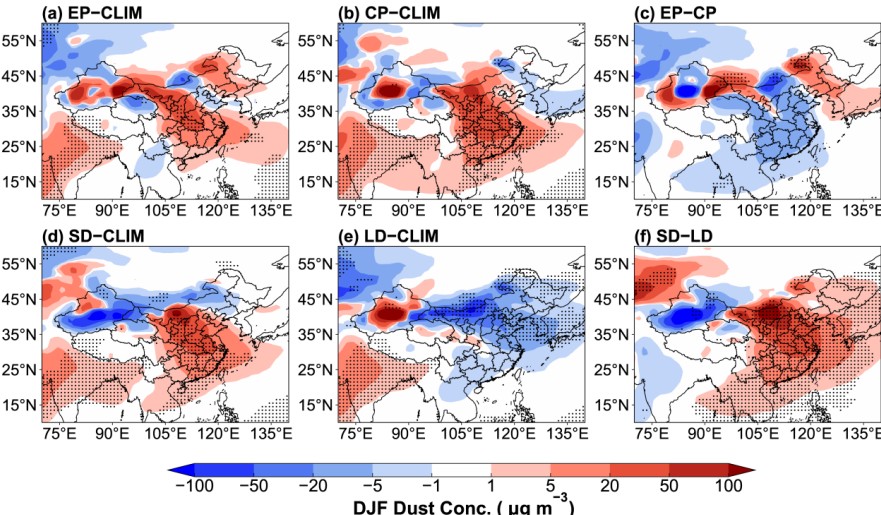

**Figure 2.** Composite differences in DJF mean near-surface dust concentrations (μg m⁻³) between
EP and CLIM in (a), CP and CLIM in (b), EP and CP in (c), SD and CLIM in (d), LD and CLIM in
(e), and SD and LD in (f). The stippled areas indicate statistical significance with 90% confidence
from a two-tailed T-test.





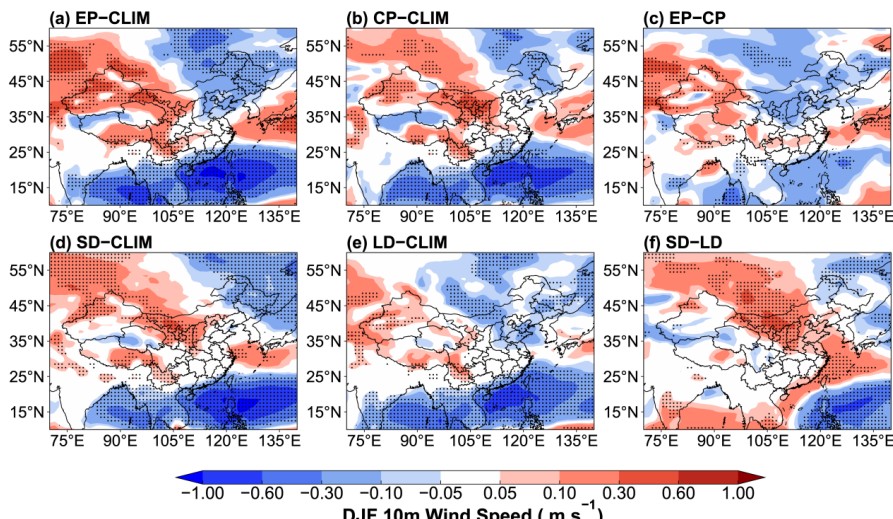

**Figure 3.** Composite differences in DJF mean 10-m wind speed (m s⁻¹) between EP and CLIM in (a), CP and CLIM in (b), EP and CP in (c), SD and CLIM in (d), LD and CLIM in (e), and SD and LD in (f). The stippled areas indicate statistical significance with 90% confidence from a two-tailed T-test.






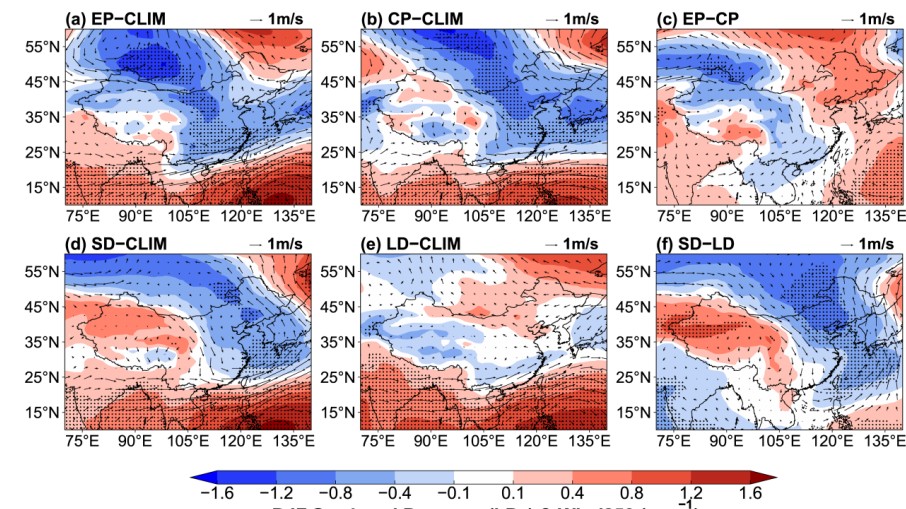


**Figure 4.** Composite differences in DJF mean sea level pressure (SLP, shaded; units: hPa) and winds
at 850 hPa (WIND850, vector; units: m s$^{-1}$) between EP and CLIM in (a,), CP and CLIM in (b), and
EP and CP in (c), SD and CLIM in (d), LD and CLIM in (e), and SD and LD in (f). The stippled
areas indicate statistical significance with 90% confidence from a two-tailed T-test.

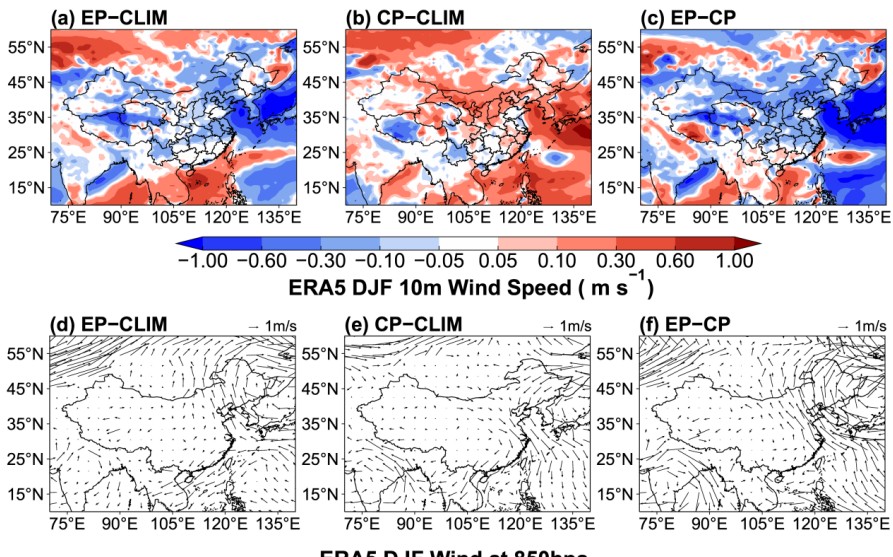

**Figure 5.** Composite differences in DJF mean 10-m wind speed (m s⁻¹) (top panels) and wind at 850 hPa (vector; units: m s⁻¹) (bottom panels) between 2006/07 EP El Niño and climatological mean (1950–2017) in (a, d), 2014/15 CP El Niño and climatological mean in (b, e), and 2006/07 EP El Niño and 2014/15 CP El Niño in (c, f) from the EAR5 reanalysis data. The data were detrended over 1950–2017.

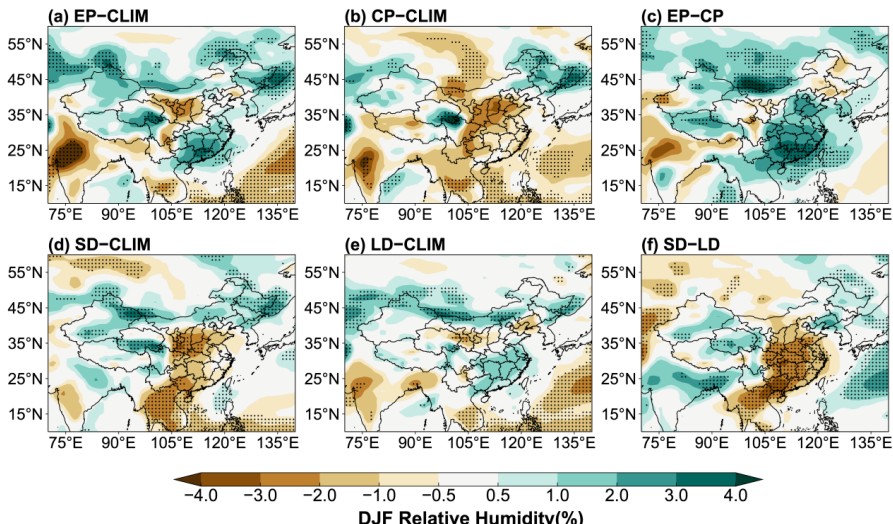

**Figure 6.** Composite differences in DJF mean relative humidity (units: %) between EP and CLIM
in (a), CP and CLIM in (b), and EP and CP in (c), SD and CLIM in (d), LD and CLIM in (e), and
SD and LD in (f). The stippled areas indicate statistical significance with 90% confidence from a
two-tailed T-test.





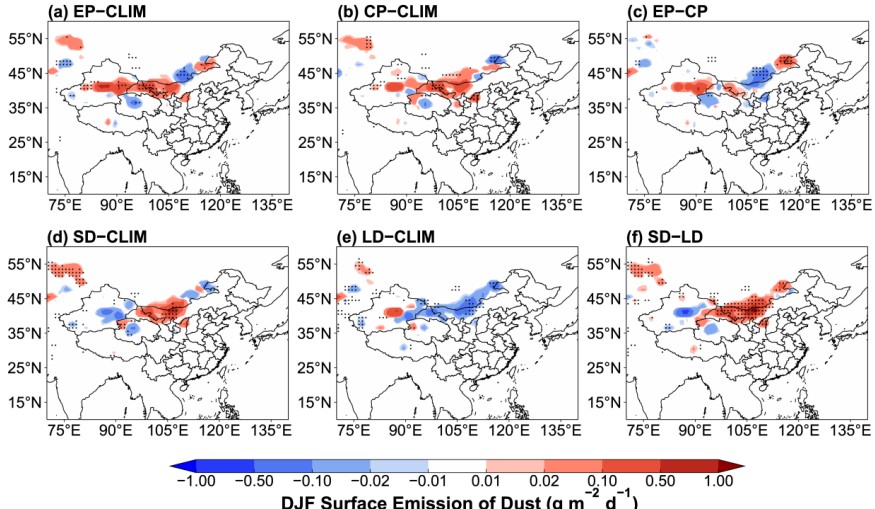

627

**Figure 7.** Composite differences in DJF mean dust emissions (g m$^{-2}$ d$^{-1}$) between EP and CLIM in

(a), CP and CLIM in (b), EP and CP in (c), SD and CLIM in (d), LD and CLIM in (e), and SD and

LD in (f). The stippled areas indicate statistical significance with 90% confidence from a two-tailed

T-test.

632


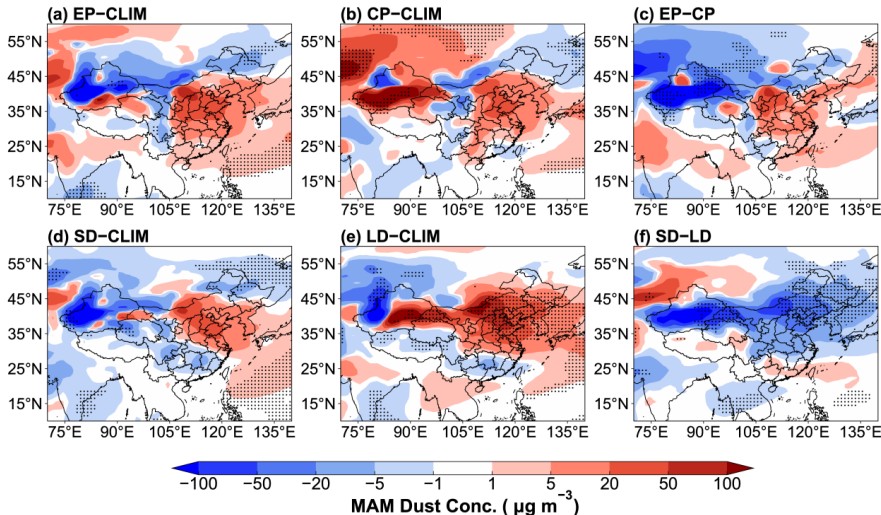

**Figure 8.** Composite differences in MAM mean near-surface dust concentrations (μg m⁻³) between
EP and CLIM in (a), CP and CLIM in (b), EP and CP in (c), SD and CLIM in (d), LD and CLIM in
(e), and SD and LD in (f). The stippled areas indicate statistical significance with 90% confidence
from a two-tailed T-test.






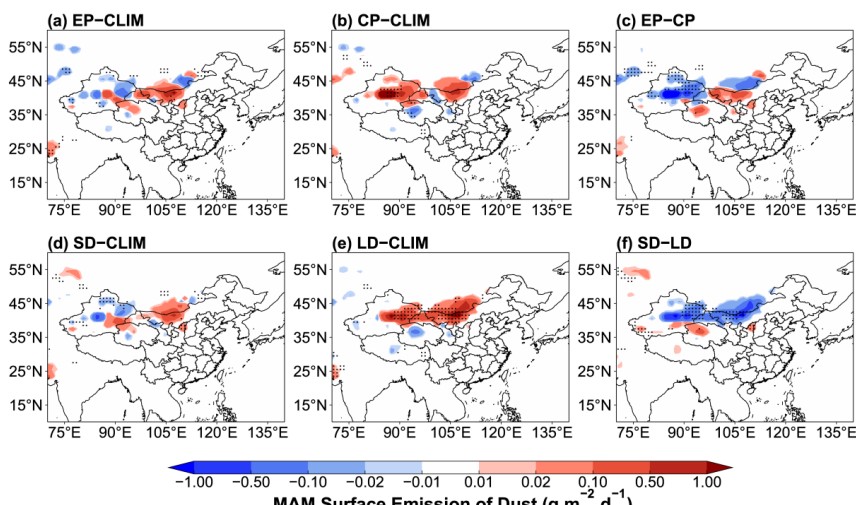


**Figure 9.** Composite differences in MAM mean dust emissions (g m⁻² d⁻¹) between EP and CLIM
in (a), CP and CLIM in (b), EP and CP in (c), SD and CLIM in (d), LD and CLIM in (e), and SD
and LD in (f). The stippled areas indicate statistical significance with 90% confidence from a two-
tailed T-test.