# Peer review of "Dust pollution in China affected by different spatial and"

_Atmospheric Chemistry and Physics, 2022_

## Author Comment (AC1)

**Manuscript # acp-2022-355**

**Responses to Referee #1**

*This study analyzed the impacts of different types of El Niño on dust activities over China using E3SMv1 model simulation. The authors showed that El Niño causes changes in boreal winter (DJF) dust concentrations over China by modulating wind speed and relative humidity near the dust sources (e.g., Gobi desert). The impacts on boreal spring (MAM) dust concentrations are statistically unclear. The influences of different types of El Niño are discussed. Overall, the paper is well organized, helpful and appropriate for publication in ACP. I may recommend publication of this manuscript after the following comments are addressed.*

We thank the reviewer for all the insightful comments. Below, please see our point-by-point response (in blue) to the specific comments and suggestions and the changes that have been made to the manuscript, in an effort to take into account all the comments raised here.

*Specific comments:*
*1. The results might be sensitive to the model selected (i.e., E3SMv1). The authors pointed out that there are model biases in simulating dust emissions (Section 2.3). However, explanation on how these biases may affect the key results (e.g., as shown in Figure 2) is not clearly discussed. The discussion given in Section 4 might be too short. Further, is it possible to suggest that these biases are associated with the model biases in simulating humidity or near surface wind?*

Response:

Thanks for the suggestion. Compared to the observations, E3SMv1 overestimates AOD near some desert source regions and the bias is even larger in its predecessor CAM5. The bias is partly related to the dust treatment in the model that dust is emitted into a shallow model bottom layer in E3SMv1 for increased model vertical resolution (Wang et al., 2020). In addition, stronger 10-m wind speed simulated by the model compared to the observation (Fig. S3) also contributes to the higher dust loading. We have now added these explanations in the model evaluation section.

In Section 4, we have also added some discussion related to the bias of dust loading, as "This high bias of dust loading near the dust source regions are related to the dust treatment in the model, dust parameterization and stronger winds in model than observations. The low bias of long-range transport of dust is due to the strong dust deposition considering that dust is emitted in the shallow model bottom layer in the model. Therefore, the estimate of El Niño impact on dust emissions and

concentrations are likely to be overestimated near the source regions, but impact from changes in large-scale circulation related to El Niño on dust transport is possibly underestimated."

[Figure]

**Figure S3.** Spatial distributions of DJF mean 10-m wind speed (m s⁻¹) (top panels) and relative humidity (units: %, bottom panels) from ERA5 over 1950–2020 in (a) and (c) and CLIM experiment in (b) and (d), respectively.

*2. I am wondering if the current model can provide the output of dust deposition and if the analysis of El Niño impacts on dust deposition is necessary.*

Response:

Thanks for your insightful suggestion. The model can provide the output of dust deposition and we have added Fig.S6 and Fig.S11 to show the dust wet deposition during different types El Niño. "Furthermore, a reduced DJF precipitation during both EP and CP El Niño events (Fig. S5) should weaken the wet removal of dust from the atmosphere in northern China. However, only insignificant decreases in wet deposition appear in part of northern China and significant increases in wet deposition are located in central and southern China related to increases in dust loading during EP

and CP El Niño events (Fig. S6). It suggests that El Niño impact on dust concentrations is mainly through changing the emission and transport of dust rather than the scavenging in winter." "Along the transport pathway, the weakened precipitation (Fig. S10) partly reduces the dust wet removal (Fig. S11), leading to the strong increase in MAM dust concentration over northern China during the LD El Niño. However, this effect is largely overwhelmed by the increased dust wet removal due to the emission-induced increase in dust concentrations." We have added these descriptions in the revised manuscript.

[Figure]

**Figure S6.** Composite differences in DJF mean dust wet deposition (mg m⁻² d⁻¹) between EP and CLIM in (a), CP and CLIM in (b), EP and CP in (c), SD and CLIM in (d), LD and CLIM in (e), and SD and LD in (f). The stippled areas indicate statistical significance with 90% confidence from a two-tailed T-test.

[Figure]

**Figure S11.** Composite differences in MAM mean dust wet deposition (mg m$^{-2}$ d$^{-1}$) between EP and CLIM in (a), CP and CLIM in (b), EP and CP in (c), SD and CLIM in (d), LD and CLIM in (e), and SD and LD in (f). The stippled areas indicate statistical significance with 90% confidence from a two-tailed T-test.

*3. Line 185: It is unclear about the period of the sensitivity experiments. Why did the authors chose 13 years only? I supposed that the period of the sensitivity experiment for SD El Niño should be longer than LD El Niño, as there are more SD El Niño events (Lines 159-160).*

Response:

As we described in Sec.2.3, sensitivity simulations are driven by the monthly SST representing the composite of El Niño events, which is generated through adding the mean monthly SST anomalies from Jul$^{0}$ to Jun$^{1}$ of the different types of El Niño events to the climatological SST between 60°S and 60°N. The monthly SST anomalies are the averages for all SD/LD/EP/CP El Niño events. Each of these 13 years are driven by the monthly SST anomalies repeating for the same year, which is used to reduce the model internal variability. The first 3 years are the model spin-up and the last 10 years are used for analysis.

*Technical corrections:*
*1. Lines 23-24: This sentence needs to be rephrased.*
Response:

We have revised the sentence as follows. "Dust is an important aerosol affecting air quality in China in winter and spring seasons. Dust in China is potentially influenced by the interannual climate variability associated with El Niño."

*2. Line 26 and others: Do you mean 'boreal winter'?*

Response:

Added.

*3. Line 26: 'an' → 'a'*

Response:

Revised.

*4. Lines 79-80: This reference (https://doi.org/10.5194/acp-22-5253-2022) might be helpful to reinforce the statement of El Niño impacts on dust activities. For example, the El Niño‐Southern Oscillation (ENSO) shows causal impacts on dust emission over the northwestern China and wetdust deposition over the eastern China. In addition, there is ENSO impacts on dust concentrations over the southern and western China.*

Response:

Thanks for providing the latest relevant reference, which are of great help to our statement of El Niño impacts on dust. We have added this reference here (Le and Bae, 2022).

*5. Line 183: If the results are based on the ensemble mean, it should be stated clearly.*

Response:

The results are based on the ensemble mean. We have now added this statement.

*6. Line 592: 98% or 90%? It is more common to use 99% or 95% significance level, instead of 98%.*

Response:

Thanks for the suggestion. We have revised the significance level to 99%.

Reference:

Le, T. and Bae, D.-H.: Causal influences of El Niño‐Southern Oscillation on global dust activities, Atmos. Chem. Phys., 22, 5253‐5263, https://doi.org/10.5194/acp-22-5253-2022, 2022.

---

## Author Comment (AC2)

**Manuscript # acp-2022-355**

**Responses to Referee #2**

*Comments and suggestions:*

*This study examines the impacts of different types of El Nino events on dust pollution in China based on an earth system model. According to the model simulation, the authors suggested that dust concentrations during the CP El Nino are much higher in northern China than during the EP El Nino. The short duration (SD) El Nino increases winter dust concentration over northern China, while there is a decrease in dust concentration during the long duration (LD) El Nino. In general, the topic of this study is interesting. However, current version of the manuscript at least needs a major revision. My comments are shown as follows.*

We thank the reviewer for all the insightful comments. Below, please see our point-by-point response (in blue) to the specific comments and suggestions and the changes that have been made to the manuscript, in an effort to take into account all the comments raised here.

*1. This study examines the impacts of different types of El Nino events (EP, CP, SD, LD) on the dust concentration over China only based on one model simulation. Can the different impacts of the four types of El Nino event on the dust concentration over China can be obtained in the observations?*

Response:

In this study, the anomalies in atmospheric circulation during different types of El Niño events are compared with reanalysis data. It confirms that the model can simulate the anomalies in the atmospheric circulation, which dominates the changes in dust distributions. We have also added Fig. S3 below to compare the modeled climatological wind speed and relative humidity with observations.

The dust concentrations are evaluated by comparing modeled concentrations with spring $PM_{10}$ concentrations over 2015–2021. The dust loading is also evaluated by comparing modeled dust optical depth with that derived from satellite data over 2001–2020. However, the anomalies of dust concentrations were not compared with observations. This is because that dust is jointly influenced by many factors in the observation other than El Niño, such as Mongolian cyclone, sea ice in the Barents Sea, sea surface temperature in Atlantic Ocean, Arctic Oscillation, and human activities (Fan et al 2016, 2018; Mao et al., 2011; Wang et al., 2021; Xiao et al., 2015; Yin et al., 2021), while this study presents the "pure" effects of El Niño on dust using an Earth system model. In addition, $PM_{10}$ is strongly influenced by other anthropogenic aerosols over eastern China, especially in hazy

winter. The comprehensive understanding of the impacts from different types of El Niño events on dust in China requires a longer-term observation with sufficient spatial coverage. We have added these discussions in the revised manuscript.

[Figure]

**Figure S3.** Spatial distributions of DJF mean 10-m wind speed (m s⁻¹) (top panels) and relative humidity (units: %, bottom panels) from ERA5 over 1950–2020 in (a) and (c) and CLIM experiment in (b) and (d), respectively.

*2. The obtained results of this study are only based on one model simulation. Many studies have demonstrated that impact of ENSO on extratropical atmospheric circulation and climate variation over East Asia are strongly model-dependent. It cannot confirm the robustness of the results obtained in this study only based on one model simulation.*

Response:

The simulated anomalies in atmospheric circulation were compared with reanalysis data and it turns out that E3SMv1 model can simulate the responses in atmospheric circulation to El Niño forcing. We agree with the reviewer that results from one model is not representative. We have added the limitation in the discussion section, as "Also, results from a single model with relative short simulations may not be representative and may not well remove the internal atmospheric variability, which can be further investigated by conducting large ensemble and longer simulations using multi-models."

*3. Lines 183-184: 3 ensemble and the last 10 years are used to analysis. It should be mentioned that there exist a large internal variability over extratropics. Thus, 3 ensemble and 10 years mean cannot well remove the internal atmospheric variability.*

Response:

The internal variability of the model could affect the results, but most of our results passed a two-tailed t-test at the 90% confidence level, indicating that the model response to different types of El Niño events outweighs the effect of the internal variability of the model. Notwithstanding, we have added the potential limitation in the revised manuscript, as "Also, results from a single model with relative short simulations may not be representative and may not well remove the internal atmospheric variability, which can be further investigated by conducting large ensemble and longer simulations using multi-models."

*4. From. Fig. 4, it shows that there exist large differences in the atmospheric anomalies over East Asia related to the four types of El Nino. First, what are the mechanisms for the formations of the atmospheric anomalies induced by the different types of El Nino. Second, what are the factors for the differences of atmospheric anomalies generated by EP and CP El Nino (SD and LD El Nino)?*

Response:

Although the mechanisms causing the atmospheric anomalies related to different types of El Niño and their differences are out of the scope of this study, knowing these mechanisms can help to understand the results. The EP El Niño is associated with basin-wide thermocline and surface wind variations and shows a strong teleconnection with the tropical Indian Ocean. In contrast, the CP El Niño appears less related to the thermocline variations and may be influenced more by atmospheric forcing. It has a stronger teleconnection with the southern Indian Ocean (Kao and Yu, 2009). Base on this, many studies have found that EP and CP El Niño cause different atmospheric circulation. Yuan and Yang (2012) pointed that the East Asia winter monsoon is weaker (stronger) during the winter of EP (CP) El Niño. Yu et al. (2020) noted that there are obvious southerly (northerly and northwesterly) wind anomalies at the middle to lower troposphere over eastern China during the winter of EP (CP) El Niño. These are consistent with the results in our study. The thermocline during LD El Niño mature phase is relatively shallower than that of SD El Niño. Moreover, the SST anomalies of SD El Niño are larger than LD El Niño, and the different depths of the thermocline indicates large differences in the recharged energy going into the eastern Pacific, which may lead to different oceanic and atmospheric conditions during their decaying periods (Guo and Tan, 2018). As

for EP and CP El Niño, the SST anomaly centers are located in the eastern and central equatorial Pacific, respectively. The different locations of the SSTA generate differences in atmospheric circulation anomalies. As for SD and LD El Niño, the locations of the SST anomalies are similar, but LD El Niño has a longer duration and weaker intensity, thus generating different atmospheric circulations. Due to these mechanisms are complex and out of the scope of our study, we have added these detailed descriptions in the supplementary material (Text S1).

*5. Lines 218-222: From Fig. 2c, the difference of dust concentrations over central-eastern China are weak and statistically insignificant. Hence, you cannot conclude that dust concentrations increase more significantly over central-eastern China. In addition, actually, from Fig. 2, differences in the dust concentrations between CP and EP are mostly insignificant in China. The related conclusions you mentioned are incorrect.*

Response:

Thanks for the point. We have revised these descriptions as follows. "During CP El Niño relative to the climatological mean, dust concentrations increase more significantly over central-eastern China, with the increases of 20–50 µg m$^{-3}$, 5–20 µg m$^{-3}$ higher than that during EP El Niño relative to the climatological mean. The large increase during CP El Niño relative to the climatological mean is also more widespread than that during EP El Niño relative to the climatological mean. Compared to CP El Niño, dust concentration over central-eastern China decreased slightly during the EP El Niño, but the changes are mostly insignificant." Other relevant conclusions in the manuscript have been modified accordingly.

*6. Lines 276-285: A comparison of Fig. 5 and Fig. 4 indicate that the simulated atmospheric circulation anomalies over East Asia show notably different with those in the observations. How can you say they are similar? In addition, the variables shown in Fig. 5 should be similar to those shown in Fig. 4. For example, SLP anomalies should be shown. In addition, the composites for the SD and LD El Nino events should also be shown in Fig. 5.*

Response:

We have now added the sea level pressure anomalies in Fig.5 comparing atmospheric circulation between EP and CP Niño events. As we illustrated in the manuscript, our previous work has confirmed the ability of E3SM in reproducing the atmospheric circulation during SD and LD El Niño events using the same SD and LD simulations (Zeng et al, 2021), so we did not repeat the figure in this study.

From Fig.4 and Fig.5, the simulated atmospheric circulation anomalies over dust source region and central-eastern China are consistent with the reanalysis data (i.e., anomalous southerly winds during EP El Niño and anomalous northwesterly during CP El Niño). These indicate that the model can roughly reproduce the atmospheric circulation features during different types of El Niño over central-eastern China. However, the simulated wind fields and the reanalysis data differ in other regions. This is because the atmospheric circulation is influenced by many factors other than El Niño in the observation, while the simulation only considers the influence of El Niño, so the results from simulation and observation are not fully consistent.

We have revised our descriptions as "It suggests that the atmospheric circulation features over central-eastern China during different types of El Niño are roughly captured by the model. However, we note that there are notably differences in atmospheric circulation over many regions of East Asia. It can be partly attributed to the model bias in reproducing the atmospheric responses to El Niño. The observations can also be induced by other climate factors besides El Niño, leading to a potential inconsistency in El Niño impact between model and observation."

[Figure]

**Figure 5.** Composite differences in DJF mean 10-m wind speed (m s⁻¹) (top panels) and sea level pressure (SLP, shaded; units: hPa) and wind at 850 hPa (WIND850, vector; units: m s⁻¹) (bottom panels) between 2006/07 EP El Niño and climatological mean (1950–2017) in (a, d), 2014/15 CP El Niño and climatological mean in (b, e), and 2006/07 EP El Niño and 2014/15 CP El Niño in (c, f) from the EAR5 reanalysis data. The data were detrended over 1950–2017.

*Minors:*

*1. Line 81: the under-->delete the*

Response:

Revised.

*2. Lines 139-140: Nino3.4 SST index is defined as area-mean SST anomalies in the Nino3.4 region.*

Response:

Revised.

*3. Definition of the EP and CP events: You should note that there also exist mixed El Nino event.*

Response:

Added.

*4. The years of CP, EP, SD and LD El Nino events should be shown in a Table.*

Response:

We have added Table S1 to show the years of EP, CP, SD and LD El Niño events.

References:

Fan, K., Xie, Z., and Xu, Z.: Two different periods of high dust weather frequency in northern China, Atmos. Ocean. Sci. Lett., 9, 263–269, https://doi.org/10.1080/16742834.2016.1176300, 2016.

Fan, K., Xie, Z., Wang, H., Xu, Z., and Liu, J.: Frequency of spring dust weather in North China linked to sea ice variability in the Barents Sea, Clim. Dyn., 51, 4439–4450, https://doi.org/10.1007/s00382-016-3515-7, 2018.

Guo, Y., and Tan, Z.: Westward migration of tropical cyclone rapid-intensification over the Northwestern Pacific during short duration El Nino, Nat. Commun., 9, 1507, https://doi.org/10.1038/s41467-018-03945-y, 2018.

Kao, H., and Yu, J.: Contrasting Eastern-Pacific and Central-Pacific Types of ENSO, J. Clim., 22, 615–632, https://doi.org/10.1175/2008JCLI2309.1, 2009.

Mao, R., Gong, D., Bao, J., and Fan, Y.: Possible influence of Arctic Oscillation on dust storm frequency in North China, J. Geogr. Sci. 21, 207 – 218, https://doi.org/10.1007/s11442-011-0839-4, 2011.

Wang, S., Yu, Y., Zhang, X., Lu, H., Zhang, X., and Xu, Z.: Weakened dust activity over China and Mongolia from 2001 to 2020 associated with climate change and land-use management,

Environ. Res. Lett., 16, 124056, https://doi.org/10.1088/1748-9326/ac3b79, 2021.

Xiao, D., Li, Y., Fan, S., Zhang, R., Sun, J., and Wang, Y.: Plausible influence of Atlantic Ocean SST anomalies on winter haze in China, Theor. Appl. Climatol., 122, 249-257, https://doi.org/10.1007/s00704-014-1297-6, 2015.

Yin, Z., Wan, Y., Zhang, Y., and Wang, H.: Why super sandstorm 2021 in North China?, Natl. Sci. Rev., https://doi.org/10.1093/nsr/nwab165, 2021.

Yu, X., Wang, Z., Zhang, H., He, J., and Li, Y.: Contrasting impacts of two types of El Niño events on winter haze days in China's Jing-Jin-Ji region, Atmos. Chem. Phys., 20, 10279–10293, https://doi.org/10.5194/acp-20-10279-2020, 2020.

Yuan, Y., and Yang, S.: Impacts of Different Types of El Niño on the East Asian Climate: Focus on ENSO Cycles, J. Clim., 25, 7702–7722, https://doi.org/10.1175/JCLI-D-11-00576.1, 2012.

Zeng, L., Yang, Y., Wang, H., Wang, J., Li, J., Ren, L., Li, H., Zhou, Y., Wang, P., and Liao, H.: Intensified modulation of winter aerosol pollution in China by El Niño with short duration, Atmos. Chem. Phys., 21, 10745–10761, https://doi.org/10.5194/acp-21-10745-2021, 2021.

---

## Referee Report (RR1)

Thanks for the author's effort in revising the manuscript. However, most comments were not well addressed.

First, the authors said the E3SMv1 can reproduce the atmospheric anomalies related to the two types of El Nino events in the observations. However, I cannot find the evidences. Actually, the authors only show the climatological distributions of DJF mean 10-m wind speed and relative humidity (Figure S3).

Second, as indicated in my previous comment, the results should be sensitive to the selected model. We cannot confirm the robustness of the results. We can obtain different conclusion if using other climate models. Actually, previous studies indicated that El Nino event cannot lead to notable climate and atmospheric anomalies over the regions to the north of China.

Third, the authors did not explain the mechanisms for the differences of the atmospheric anomalies over North China between different types of El Nino.

Fourth, as indicated in previous comment, in my view, the simulated atmospheric circulation anomalies over East Asia show notably different with those in the observations. I did not think they are similar.

Fifth, I do not think 3 ensemble and 10 years mean can well remove the internal atmospheric variability in mid-latitude regions, although the authors suggested that the model response to different types of El Niño events outweighs the effect of the internal variability of the model.

Deser C, Phillips AS, Alexander MA, Smoliak BV (2014) Projecting North American climate over the next 50 years: uncertainty due to internal variability. J Clim 27(6):2271–2296.

---

## Author Response (AR2)

**Manuscript # acp-2022-355**

**Responses to Referee #3**

*Thanks for the author's effort in revising the manuscript. However, most comments were not well addressed.*

We thank the reviewer for all the insightful comments. Below, please see our point-by-point response (in blue) to the specific comments and suggestions and the changes that have been made to the manuscript, in an effort to take into account all the comments raised here.

*Specific comments:*

*1. First, the authors said the E3SMv1 can reproduce the atmospheric anomalies related to the two types of El Nino events in the observations. However, I cannot find the evidences. Actually, the authors only show the climatological distributions of DJF mean 10-m wind speed and relative humidity (Figure S3).*

The atmospheric anomalies related to the different spatial types of El Nino events in the observations were also given in the manuscript to compare with those from simulations. For example, Fig.6 and Fig. S4 show the anomalies of relative humidity during different types of El Niño events. Figs. 3 and 4 show simulated 10-m wind speed, winds at 850 hPa and sea level pressure, while the same variables in observations are given in Fig. 5 (EP/CP) and our previous work (Zeng et al., 2021 for SD/LD 850 hPa winds, as shown below). These indicate E3SMv1 can roughly capture atmospheric anomalies in the observations over central-eastern in China. We have revised the figure description to emphasize which one is from observations and correct the description that E3SMv1 can "roughly capture" the atmospheric anomalies over central-eastern China rather than "reproduce" the anomalies. We also note that there are notably differences in atmospheric circulation over many regions of East Asia. It can be partly attributed to the model bias in reproducing the atmospheric responses to El Niño. The observations can also be induced by other climate factors besides El Niño, leading to a potential inconsistency in El Niño impact between model and observation.

[Figure]

**Figure 4.** Composite differences in DJF mean sea level pressure (SLP, shaded; units: hPa) and winds at 850 hPa (WIND850, vector; units: m s-1) between SD and CLIM in (d), LD and CLIM in (e), and SD and LD in (f). The stippled areas indicate statistical significance with 90% confidence from a two-tailed T-test.

[Figure]

**Figure in Zeng et al. (2021).** Composite differences in DJF mean winds at 850 hPa (m s$^{-1}$) between 2015/2016 SD El Niño and climatological mean (1950-2017) (a), 1986/1987 LD El Niño and climatological mean (b), and 2015/2016 SD El Niño and 1986/1987 LD El Niño (c) from the EAR5 reanalysis data. The data were detrended over 1950-2017.

*2. Second, as indicated in my previous comment, the results should be sensitive to the selected model. We cannot confirm the robustness of the results. We can obtain different conclusion if using other climate models. Actually, previous studies indicated that El Nino event cannot lead to notable climate and atmospheric anomalies over the regions to the north of China.*

We agree with the reviewer that the results are potentially model dependent. But we also notice that some previous studies have shown that changes in tropical sea surface temperature (SST) can lead to notable climate and atmospheric anomalies over northern China. For example, Liu et al. (2022) used the Community Atmosphere Model version 4 (CAM4) to investigate the impact of SST on dust activities in the Gobi Desert and North China and they also noted that tropical Pacific SST variability resulted in the important change in boreal spring dust activity frequency in the Gobi Desert. Moreover, Jung et al. (2022) also pointed out the important modulation of $PM_{10}$ in East Asia by the Madden-Julian Oscillation (MJO), an important meteorological phenomenon in the tropics. Le et al. (2022) and Li et al. (2021) both reported the effect of dust over northwestern China influenced by

ENSO. Due to the limitation of computational resources, we cannot repeat the simulations using different models. Therefore, we added the caveat in the discussion as "results from a single model with relative short simulations may not be representative and may not well remove the internal atmospheric variability (Deser et al., 2014), which can be further investigated by conducting large ensemble and longer simulations using multi-models."

*3. Third, the authors did not explain the mechanisms for the differences of the atmospheric anomalies over North China between different types of El Nino.*

The mechanisms for the differences of the atmospheric anomalies between different types of El Nino have been illustrated in many studies. Western North Pacific anomalous anticyclones (WNPAC), which occur during both EP and CP El Niño events, have been proved as a crucial system that links El Niño and East Asian climate (Li et al., 2017). The anomalous southwesterlies at the north of WNPAC transport moisture to southern China, which can block the prevailing northerlies over central-eastern China in winter and weaken the East Asian winter monsoon (Yuan and Yang, 2012). EP El Niño exerts larger meteorological changes over southern China than CP El Niño due to a stronger WNPAC (Jiang and Li, 2022; Kim et al., 2021). Therefore, the anomalous northerlies over the Gobi Desert and central China are hindered and weaker during EP El Niño than CP El Niño (Fig. 4). SD El Niño has a relatively deeper thermocline during its mature phase than LD El Niño and numerous ocean heat can be transported from the eastern Pacific to the South China Sea and the Western Philippine Sea during SD El Niño (Guo and Tan, 2018). The transmitted ocean heat leads to anomalous warming of the North Pacific SST, a smaller-than-normal tilt of the East Asian trough, a weakening of the mid-latitude westerly flow in front of the trough, and anomalous northerly winds along the trough line of the subtropical trough, along with reduced precipitation (Wang et al., 2009). These favor dust emission and transport from north to south during SD El Niño. We have added the detailed description in Sec. 4 of our revised manuscript.

*4. Fourth, as indicated in previous comment, in my view, the simulated atmospheric circulation anomalies over East Asia show notably different with those in the observations. I did not think they are similar.*

They do show some differences in many regions of East Asia and we did not expect or say they are similar over the whole East Asia. Although the simulated atmospheric circulation anomalies over East Asia show notably different from the observations, the anomalies over northern China, especially in the dust source region, are similar to the observations, which are the key region our

manuscript concerns. Therefore, in the last revision, we have added a notice that "It suggests that the atmospheric circulation features over central-eastern China during different types of El Niño are roughly captured by the model. However, we note that there are notably differences in atmospheric circulation over many regions of East Asia. It can be partly attributed to the model bias in reproducing the atmospheric responses to El Niño. The observations can also be induced by other climate factors besides El Niño, leading to a potential inconsistency in El Niño impact between model and observation."

*5. Fifth, I do not think 3 ensemble and 10 years mean can well remove the internal atmospheric variability in mid-latitude regions, although the authors suggested that the model response to different types of El Niño events outweighs the effect of the internal variability of the model.*
*Deser C, Phillips AS, Alexander MA, Smoliak BV (2014) Projecting North American climate over the next 50 years: uncertainty due to internal variability. J Clim 27(6):2271–2296.*

We agree with the reviewer that the simulations are relatively short. Due to the limitation of resources and the much more complexity of E3SM than its predecessor CESM1, it is difficult to prolong all simulations for all ensembles. We have tested the SD/LD results by prolonging one ensemble simulations for additional 10 years and the spatial pattern of the dust response does not have a large change over China, except over parts of northeastern China in LD simulation. We also have a caveat that "results from a single model with relative short simulations may not be representative and may not well remove the internal atmospheric variability (Deser et al., 2014), which can be further investigated by conducting large ensemble and longer simulations using multi-models."

[Figure]

**Figure A.** Composite differences in DJF mean near-surface dust concentrations (µg m⁻³) between SD/LD and CLIM for 10-year (top) and 20-year (bottom) simulations with one ensemble. The stippled areas indicate statistical significance with 90% confidence from a two-tailed T-test.

Reference:

Deser, C., Phillips, A. S., Alexander, M. A., and Smoliak, B. V., Projecting North American climate over the next 50 years: uncertainty due to internal variability. J. Clim., 27, 2271–2296, 2014, https://doi.org/10.1175/JCLI-D-13-00451.1.

Guo, Y., and Tan, Z.: Westward migration of tropical cyclone rapid-intensification over the Northwestern Pacific during short duration El Nino, Nat. Commun., 9, 1507, https://doi.org/10.1038/s41467-018-03945-y, 2018.

Jiang, Z. and Li, J.: Impact of eastern and central Pacific El Niño on lower tropospheric ozone in China, Atmos. Chem. Phys., 22, 7273–7285, https://doi.org/10.5194/acp-22-7273-2022, 2022.

Jung, M., Son, S., Kim, H., and Chen, D.: Tropical modulation of East Asia air pollution. Nat. Commun., 13, 5580, https://doi.org/10.1038/s41467-022-33281-1, 2022.

Kim, J., Chang, T., Lee, C., and Yu, J.: On the Varying Responses of East Asian Winter Monsoon to Three Types of El Niño: Observations and Model Hindcasts, J. Clim., 34(10), 4089-4101, https://doi.org/10.1175/JCLI-D-20-0784.1, 2021.

Le, T. and Bae, D.-H.: Causal influences of El Niño-Southern Oscillation on global dust activities,

Atmos. Chem. Phys., 22, 5253-5263, https://doi.org/10.5194/acp-22-5253-2022, 2022.

Li, J., Garshick, E., Huang, S., and Koutrakis, P.: Impacts of El Nino-Southern Oscillation on surface dust levels across the world during 1982-2019, Sci. Total Environ., 769, 144566, https://doi.org/10.1016/j.scitotenv.2020.144566, 2021.

Li, T., Wang, B., Wu, B., Zhou, T., Chang, C. P., and Zhang, R.: Theories on formation of an anomalous anticyclone in western North Pacific during El Niño: A review, J. Meteorol. Res., 31, 987–1006, https://doi.org/10.1007/s13351-017-7147-6, 2017.

Liu, G., Li, J., Jiang, Z., and Li, X.: Impact of sea surface temperature variability at different ocean basins on dust activities in the Gobi Desert and North China, Geophys. Res. Lett., 49, e2022GL099821, https://doi.org/10.1029/2022GL099821, 2022.

Wang, L., Chen, W., Zhou, W., and Huang, R.: Interannual Variations of East Asian Trough Axis at 500 hPa and its Association with the East Asian Winter Monsoon Pathway, J. Clim., 22(3), 600-614, https://doi.org/10.1175/2008JCLI2295.1, 2009.

Yuan, Y., and Yang, S.: Impacts of Different Types of El Niño on the East Asian Climate: Focus on ENSO Cycles, J. Clim., 25, 7702–7722, https://doi.org/10.1175/JCLI-D-11-00576.1, 2012.

Zeng, L., Yang, Y., Wang, H., Wang, J., Li, J., Ren, L., Li, H., Zhou, Y., Wang, P., and Liao, H.: Intensified modulation of winter aerosol pollution in China by El Niño with short duration, Atmos. Chem. Phys., 21, 10745–10761, https://doi.org/10.5194/acp-21-10745-2021, 2021.